# Biocompatibility and Degradation Behavior of Molybdenum in an In Vivo Rat Model

**DOI:** 10.3390/ma14247776

**Published:** 2021-12-16

**Authors:** Antje Schauer, Christian Redlich, Jakob Scheibler, Georg Poehle, Peggy Barthel, Anita Maennel, Volker Adams, Thomas Weissgaerber, Axel Linke, Peter Quadbeck

**Affiliations:** 1Laboratory of Experimental and Molecular Cardiology, Dresden University of Technology, Heart Center Dresden, 01307 Dresden, Germany; peggy.barthel@tu-dresden.de (P.B.); anita.maennel@tu-dresden.de (A.M.); volker.adams@tu-dresden.de (V.A.); axel.linke@tu-dresden.de (A.L.); 2Dresden Branch Lab., Fraunhofer Institute for Manufacturing Technology and Advanced Materials IFAM, Winterbergstraße 28, 01277 Dresden, Germany; christian.redlich@ifam-dd.fraunhofer.de (C.R.); jakob.scheibler@ifam-dd.fraunhofer.de (J.S.); georg.poehle@ifam-dd.fraunhofer.de (G.P.); Thomas.Weissgaerber@ifam-dd.fraunhofer.de (T.W.); peter.quadbeck@ifam-dd.fraunhofer.de (P.Q.); 3Dresden Cardiovascular Research Institute and Core Laboratories GmbH, 01099 Dresden, Germany

**Keywords:** molybdenum, bioresorbable, biocompatibility, degradation, in vivo, blood analysis, organ accumulation, histologic analysis

## Abstract

The biocompatibility and degradation behavior of pure molybdenum (Mo) as a bioresorbable metallic material (BMM) for implant applications were investigated. In vitro degradation of a commercially available Mo wire (ø250 µm) was examined after immersion in modified Kokubo’s SBF for 28 days at 37 °C and pH 7.4. For assessment of in vivo degradation, the Mo wire was implanted into the abdominal aorta of female Wistar rats for 3, 6 and 12 months. Microstructure and corrosion behavior were analyzed by means of SEM/EDX analysis. After explantation, Mo levels in serum, urine, aortic vessel wall and organs were investigated via ICP-OES analysis. Furthermore, histological analyses of the liver, kidneys, spleen, brain and lungs were performed, as well as blood count and differentiation by FACS analysis. Levels of the C-reactive protein were measured in blood plasma of all the animals. In vitro and in vivo degradation behavior was very similar, with formation of uniform, non-passivating and dissolving product layers without occurrence of a localized corrosion attack. The in vitro degradation rate was 101.6 µg/(cm^2^·d) which corresponds to 33.6 µm/y after 28 days. The in vivo degradation rates of 12, 33 and 36 µg/(cm^2^·d) were observed after 3, 6 and 12 months for the samples properly implanted in the aortic vessel wall. This corresponds with a degradation rate of 13.5 µm/y for the 12-month cohort. However, the magnitude of degradation strongly depended on the implant site, with the wires incorporated into the vessel wall showing the most severe degradation. Degradation of the implanted Mo wire neither induced an increase in serum or urine Mo levels nor were elevated Mo levels found in the liver and kidneys compared with the respective controls. Only in the direct vicinity of the implant in the aortic vessel wall, a significant amount of Mo was found, which, however, was far below the amounts to be expected from degrading wires. No abnormalities were detected for all timepoints in histological and blood analyses compared to the control group. The C-reactive protein levels were similar between all the groups, indicating no inflammation processes. These findings suggest that dissolved Mo from a degrading implant is physiologically transported and excreted. Furthermore, radiographic and µCT analyses revealed excellent radiopacity of Mo in tissues. These findings and the unique combination with its extraordinary mechanical properties make Mo an interesting alternative for established BMMs.

## 1. Introduction

The ongoing development of bioresorbable metallic materials (BMMs) perspectively renders the planned or unplanned removal of certain permanent implants unnecessary. This is facilitated by BMMs’ unique characteristic to provide structural support during tissue healing and remodeling and being subsequently dissolved by corrosion and metabolized by the restored tissues. Besides orthopedic application, BMMs are ideally suited for the load-bearing use in cardiovascular stenting. A functioning bioresorbable stent may thus prevent the persistently occurring in-stent restenosis, chronic inflammation, and late cardiovascular thromboses associated with permanent stents.

Up to date, four physiologically essential metals were investigated as basis for BMMs: magnesium (Mg), iron (Fe), zinc (Zn) [1], and molybdenum (Mo) [2]. Among these, Mg alloys are the most investigated due to their excellent biocompatibility and resorbability properties [1,3]. The first BMM implants that have been granted market authorization were made of bioresorbable Mg alloys, e.g., the Magnezix^®^ bone pins and screws (Syntellix AG, Hannover, Germany) or the Magmaris^®^ stent system (Biotronik AG, Germany). However, their generally fast and nonuniform corrosion accompanied by hydrogen gas evolution alongside low mechanical strength and Young’s modulus limits the broader use of Mg alloys. A clinical application of high-strength resorbable Fe materials has been hindered by slow in vivo degradation due to formation of insoluble corrosion products [4]. However, there are some recent promising results for polymer-coated nitrided iron stents with enhanced degradation rate and corrosion product decomposition [5,6]. Bioresorbable Zn materials, whose mechanical properties and degradation rates are between those of Mg and Fe [7], are still afflicted with low ductility, strain rate-dependent mechanical properties and age-hardening effects [8].

Recently, we reported on the potential of pure Mo as a novel BMM [2,9]. Mo exhibits high mechanical strength, e.g., an ultimate tensile strength (UTS) of nearly 1400 MPa and Young’s modulus of 320 GPa in comparison to 316 L stainless steel with 340 MPa and 193 GPa [10]. Alongside an elongation to failure of up to 50% [11] and excellent fatigue behavior [12,13], this is favorable for load-bearing implant applications. In combination with in vitro dissolution rates of 24 µg/(cm^2^·d) which corresponds to approximately 10 µm/y for commercially available pure Mo after 28 days of immersion in modified Kokubo’s simulated body fluid [2], Mo appears especially attractive for small load-bearing implants like stents. This degradation rate is close to a commonly referenced benchmark value of 20 µm/y for resorbable stent material (for stent struts of 80 µm completely degrading in 2 years) [7].

The anodic dissolution reaction for molybdenum is:(1)$${\mathrm{Mo}+4H}_{2}O\;\to {\;\mathrm{MoO}}_{4}{}^{2–}{+8H}^{+}{+6e}^{–}$$
with metabolizable molybdate ions MoO_4_^2−^ being the predominant species in aqueous media at pH 7.4 [14,15]. It was shown that Mo forms a non-passivating, uniform, and thin degradation product layer [9,16] that slowly dissolves to MoO_4_^2−^ and to a lesser extent to higher molybdates by a reaction cascade from metallic Mo, over IV-, V-, and VI-valent oxides [17,18].

Excess Mo beyond the physiologically normal total values of 10–15 mg is renally excreted, which should allow for metabolic clearance of molybdate released from a degrading Mo implant without tissue accumulation [19]. The recommended dietary Mo upper intake level for humans is 0.6–2 mg/d [20]. Furthermore, no cytotoxic effects on fibroblasts and osteoblasts were found at concentrations below 119 mg/L [21]. Our previous in vitro study can be summarized as follows: human coronary endothelial (HCAEC) and coronary artery smooth muscle cells (HCASMC) did not show signs of apoptosis at exposition for up to 72 h to Mo concentrations up to 15 mg/L and 120 mg/L, respectively. This accounts for the 320-fold and 2,500-fold concentration expected for a pure Mo stent with a surface area of 2 cm^2^ based on the reported in vitro dissolution rate of 24 µg/(cm^2^·d). Furthermore, no effect on inflammatory response was found for human THP-1 monocytes exposed to Mo concentrations up to 120 mg/L. Mo and 316 L stainless steel were found to be equally colonizable with HCAEC and HCASMC. No thrombogenicity towards human thrombocytes was observed [2].

Thus, in vitro investigations of Mo indicate good biocompatibility and great potential for an application in cardiovascular or orthopedic implants. To take the next step towards qualification of Mo as a BMM, in this study, microstructure and in vitro immersion degradation behavior of a commercially available Mo wire was characterized, and it was implanted into the vessel walls of the descending aorta of 39 female Wistar rats. The wire samples were explanted after 3, 6, and 12 months, and in vivo degradation was assessed by metallography and SEM/EDX analysis. Mo concentration in urine and serum samples as well as lysed organ samples was investigated via ICP-OES analysis. The liver, kidney, brain, spleen and lung samples were also examined histologically.

## 2. Materials and Methods

### 2.1. Material Preparation

A commercially available cold-drawn and electropolished Mo wire with R = 250 µm diameter was purchased from Plansee India High Performance Materials Pvt. Ltd., Mysore, India. Impurity levels as specified by the manufacturer were as follows (in ppm): Al < 10, Cr < 20, Cu < 20, Fe < 20, Ni < 10, Si < 20, W < 300, O < 40.

For the static immersion test, the wire was cut with pincers into pieces of 280 ± 2 mm length. This corresponds to a surface area of 2.20 ± 0.01 cm^2^ and a weight of 141 ± 1 mg. The ends of the wires were blunted by grinding with 1200 grit SiC abrasive paper. The wires were bent into spirals to fit into the test containers and cleaned with ethanol in an ultrasonic bath.

The samples to be implanted in the descending aorta of Wistar rats were cut with pincers into pieces of approx. 15 mm length. Their ends were blunted by grinding with 1200 grit SiC abrasive paper. The weight of a prepared sample was 7.5 ± 0.2 mg and the length was 14.8 ± 0.3 mm, corresponding to a surface area of 11.7 ± 0.3 mm^2^.

### 2.2. Static Immersion Test

Immersion tests were performed based on ASTM-G31–72 [22]. One sample each of the Mo wire was placed in a 50 mL PVC bottle and immersed in modified corrected Kokubo’s simulated body fluid with decreased Ca amount (c-SBF–Ca) for 28 days at 37 °C with a ratio of 10 mL per cm^2^ surface area in an incubator (VWR International GmbH, Incu-Line IL 23). The method and the composition of c-SBF–Ca was described in detail previously [2]. After 28 days, the samples were removed from the medium, rinsed with distilled water and ethanol, and dried in air.

Degradation rates and dissolution rates were determined from the test. The term “degradation” means the complete amount of Mo that is changed by corrosion from metallic state to solid oxides/hydroxides and dissolved products, whilst “dissolution” refers to the amount of Mo that is actually dissolved in the electrolyte.

Degradation is derived from the radial loss *b*_deg_ of the wires that was determined by cross-sectional preparation of the corroded wires. Due to the uniform degradation, the degraded mass of Mo can be calculated from the degraded volume *V*_deg_ in the form of a hollow cylinder with the respective length of the wires *l*:(2)$$\Delta m_{\deg}=\frac{{\;V}_{\deg\cdot }\rho_{\mathrm{Mo}}}{A_{\mathrm{sample}}}=\frac{\;\pi \left(R^{2}-{\left(R-b_{\deg}\right)}^{2}\right)µm\cdot l\cdot \rho_{\mathrm{Mo}}\cdot {\;10}^{4}\;\frac{µm}{\mathrm{cm}}}{A_{\mathrm{sample}}}$$

Degradation rates *v*_deg_ can then be easily derived by dividing Δ*m*_Mo deg_ by the immersion or implantation time *t*. The degradation depth *d*_deg_ in µm/y is obtained by multiplying *v*_deg_ by 365 d/y. The influence of gradual reduction of the active metallic surface area *A*_sample_ is relatively small for small ratios of *b*_deg_ to the initial wire diameter and is therefore omitted in this study. However, for thinner wires or larger radial losses *b*_deg_ due to degradation, it must be considered.

For the assessment of dissolution, the used electrolyte was changed twice a week in alternating 4- and 3-day intervals and concentration of dissolved Mo was analyzed by inductively coupled plasma optical emission spectroscopy (ICP-OES, iCAP 6000, Thermo Fisher Scientific). From the measured Mo concentrations *c*_Mo diss,i_, the specific mass loss Δ*m*_Mo diss,i_ in mg/cm^2^ for each timepoint *t*_i_ was calculated, summed up, and plotted against the time of immersion:(3)$$\underset{i}{\sum }\Delta m_{\mathrm{diss},i}=\frac{{\;c}_{\mathrm{Mo}\;\mathrm{diss},i}\;\cdot {\;V}_{\mathrm{medium}}}{A_{\mathrm{sample}}}{=c}_{\mathrm{Mo}\;\mathrm{diss},i}\cdot \;0.01\frac{L}{{\mathrm{cm}}^{2}}$$

Dissolution rates *v*_diss_ in terms of daily mass losses (mg/(cm^2^·d)) were derived by dividing Δm_Mo,i_ by the respective timepoint *t*_i_.

### 2.3. Wire Implantation

Female Wistar rats were obtained from Janvier Labs (France) and included in the experiment at 12 weeks of age. All the rats were housed under standard conditions and maintained on normal rat chow. All the procedures were licensed (25-5131/474/26) and carried out according to the institutional animal care guidelines as regulated by the German federal law governing animal welfare.

A previously described surgical protocol [23,24] was performed with small alterations. In brief, the rats were anesthetized with an antagonizable mixture (medetomidine/midazolam/fentanyl: 0.15/2/0.005 mg/kg body weight (BW)) to adapt the anesthesia to the duration of surgery. The depilated, sterilized abdominal wall was opened and the intestine was dislodged in a moist towel to gain visibility of the abdominal aorta. The surrounding connective tissue was removed and the aorta was stretched using tweezers to interrupt blood flow. The aortic wall was pre-punctured with a sterilized 27-gauge cannula before the previously sanitized (70% ethyl alcohol) blunt wire shown in Figure 1a was inserted and advanced over a length of about 10 mm. The wire ends were carefully bent to avoid slipping in the vessel. Figure 1b shows a wire after implantation. As the outcome, both wall implants and luminal implants were realized (schematically shown in Figure 1c). After wound closure, the anesthesia was terminated by the administration of the narcotic antagonists (atipamezole hydrochloride/flumazenil/naloxone: 0.75/0.2/0.12 mg/kg BW). The rats were randomly assigned to three groups (3, 6, 12 months) with n = 13, n = 14, and n = 12, respectively. The six female Wistar rats who did not undergo surgery served as the age-matched controls.

### 2.4. Radiography and Computed Tomography (CT)

To analyze the X-ray visibility of the wires, a subset of euthanized rats was used for X-ray imaging and CT. For reference, an additional Mo wire sample was positioned on the belly of the rats. Cone beam computed tomography imaging was performed using a small animal image-guided radiotherapy [25] platform as described before [26] with the following parameters: 60 kV and 0.5 mA. CBCT reconstruction was performed using the ConeBeam software [25] and visualized using in-house developed software.

### 2.5. Explantation and Tissue Asservation

After 3, 6, or 12 months, prior to the experimental endpoint, blood samples were taken from the lateral tail vein before the rats were euthanized using thiopental (0.25 mg/kg BW). The aortae were carefully explanted together with the wires. Under stereomicroscopic view, they were photographed in the closed and the longitudinally sliced state. Urine was collected and snap-frozen in liquid nitrogen. Organs, including the kidneys, liver, spleen, lungs, and brain were preserved in paraformaldehyde (4%, PBS-buffered) for histological examination as well as snap-frozen in liquid nitrogen for further analysis of organ Mo concentrations by ICP-OES.

### 2.6. Organ Preperation and Analyses of Mo Concentration

Serum was diluted 1:5, urine—either 1:5 or 1:10 (depending on the individual urine quantity) in 100 mM Tris (Tris (hydroxymethyl) aminomethane) buffer at pH 7.4 supplemented with 2% SDS (sodium dodecyl sulfate) and 2% glycerin. All the organs to be examined, except for the aorta, were homogenized in liquid nitrogen using a mortar and pestle. Approximately 50–80 mg of the tissue powder was diluted in the buffer described above. Segments of abdominal aortae from the vicinity of the implants of approximately 5 mm length and 15 mg mass were homogenized (Precellys^®^ 24 homogenizer, Bertin Technologies, Germany) in 300 µL of Tris buffer without SDS to avoid foaming. The homogenate was incubated overnight at 4 °C, centrifuged at 2000 rpm to remove the undigested material and the supernatant was diluted 1:15 using the same buffer. Protein concentration of all the organs was determined by BCA (bicinchoninic acid) assay (Pierce, Germany) using standard protocols.

The Mo concentration in 5 mL extracts each of urine, serum, and organs prepared as described above were analyzed by means of ICP-OES. Urine and serum concentrations were calculated from the ICP-OES-derived Mo concentration and are quoted in ng/mL. Mo concentrations in the organs are provided in µg/g in correspondence to the organ sample protein mass. From this, the total Mo amount in the respective whole organs was evaluated.

### 2.7. Microstructural and Corrosion Product Characterization

The microstructure of the Mo wires as-received as well as the degradation products formed during static immersion in vitro or implantation in vivo were investigated by means of optical microscopy (OM, Reichert MEF4A) and scanning electron microscopy (SEM, Jeol JSM-IT800). The elemental composition was evaluated by means of energy dispersive X-ray analysis (EDX, Bruker XFlash Detektor 6/30). Due to the semiquantitative nature of the analysis, light elements like oxygen cannot be reliably quantified.

The surface morphology of degradation products after immersion or implantation was investigated without further preparation of the samples. Cross-sections of uncorroded and corroded Mo samples were prepared by embedding in epoxy, wet grinding using abrasive SiC paper up to 2500 grit, and polishing sequentially with a diamond suspension (3 µm and 1 µm) and an alumina suspension (0.1 µm). The samples were rinsed in water, ultrasonically cleaned in ethanol, and dried in warm air. Prior to SEM/EDX analysis, the samples were coated with carbon and contacted with conductive silver paste.

### 2.8. Blood Count and Inflammation

Red blood cells (RBC), white blood cells (WBC), and platelets (PLT) were counted using fluorescence flow cytometry (Sysmex XT-2000, Sweden) following the manufacturer’s protocol. In brief, 80 µL whole blood received out of the tail vein using a heparin-moistened syringe were diluted with 320 µL 0.9% sodium chloride for double measurement of each sample. The hand-warm samples were briefly vortexed and directly used for automated hematology analyses. The system allows counting of RBC and PLT by impedance technology while WBC are counted using multi-angle scatter separation. WBC were further analyzed for subpopulations (neutrophil granulocytes, lymphocytes, and monocytes) by means of a combination of scatter signals and a polymethine fluorescence dye signal.

For the detection of potential systemic inflammatory processes, the classic acute-phase C-reactive protein (CRP, Tina-quant C-Reactive Protein, Roche, Germany) was measured in blood plasma with Cobas c701 according to the manufacturer’s protocol. In brief, for this particle-enhanced immunoturbidimetric assay, 2 µL of blood plasma were added to 150 µL reagent 1 (Tris buffer with bovine serum albumin; preservatives), 48 µL reagent 2 (latex particles coated with anti-CRP in glycine buffer; immunoglobulins; preservative), and 24 µL H_2_O, and the formed aggregates were subsequently determined turbidimetrically.

### 2.9. Histology

Prior to staining, paraffin-embedded organ sections (4 μm) were deparaffinized (xylene and descending ethanol series) and rehydrated in H_2_O. After the respective staining, the sections were dehydrated using ascending ethanol series. The sections were either stained with hematoxylin–eosin (H&E, spleen and brain), Azan stain (spleen, liver), Masson–Goldner staining kit (kidneys), or Elastica van Gieson staining kit (lung) using standard protocols. In brief, for Azan staining, the nuclei were stained red with azocarmine while connective and muscle tissues were counter-dyed with aniline blue–orange. For Masson–Goldner staining, the nuclei were stained with iron hematoxylin while counter-staining was performed with ponceau–acid fuchsin, phosphotungstic acid–orange G, and light green SF. For Elastica van Gieson staining, the nuclei were stained with iron hematoxylin while connective and muscle tissues were counter-dyed with picric acid and acid fuchsin.

### 2.10. Statistics

Three samples were tested in the immersion corrosion tests. Systematic relative maximum errors were estimated by means of propagation of uncertainty methods. Errors were estimated to be < 1% for the high-precision method of ICP-OES mass loss measurement. Experimental mean deviations (MD) were much higher and therefore depicted as error bars (mean ± MD).

In the in vivo test, despite the relatively high number of specimens for each timepoint, the mean degradation rate was not calculated due to significant influence of the implant site. Therefore, the discussion focuses on specimens that are deemed representative, i.e., implants properly integrated in the aortic vessel wall.

For the serum, urine, and organ extract analyses via ICP-OES, each extract was analyzed three times, and a mean was calculated. The values depicted in Figure 8 are the arithmetic mean values with the mean deviations (mean ± MD) derived from those individual means.

Data of blood analyses are presented as the means ± SEM (standard error of the mean). For each timepoint, values of the treated group were compared to values of the age-matched control group using two-sided Student’s t-test of equal variances. A p-value below 0.05 was considered as statistically significant.

## 3. Results and Discussion

### 3.1. In Vitro Static Immersion Corrosion Behavior

The appearance and characteristics of the commercially available Mo wire are summarized in Figure 2. The original metallic appearance of the spiraled wire is shown on the left side in Figure 2a. The addition of potentially corrosion-enhancing dislocations in the material through shaping of the spirals was deemed insignificant considering the cold-drawn state of the wire. The typical elongated microstructure parallel to the direction of wire drawing is observable in Figure 2b. The elongated faces of the grains are exposed to the physiological medium in vitro or in vivo. The influence of the cross-sectional fine-grained microstructure shown in Figure 2c is negligible due to the small diameter of the wire. The original surface of the wire is shown in Figure 2d. Despite electropolishing, marks of the drawing process are still visible. However, the resulting increase in the surface area is deemed insignificant since there is no pronounced roughness of the wire visible in Figure 2c. This is the same condition in which the wires were implanted into the descending aortae of the Wistar rats.

When immersed in c-SBF–Ca, a gradual change of color from silver metallic to lustrous black due to the formation of degradation products was observed within the first 3 days. The appearance of the wire after 28 days is shown on the right side in Figure 2a. This color change was also observed in all the other Mo samples in our recent study [2].

A representative cross-sectional SEM image of the corroded wire after 28 days is shown in Figure 2e. Figure 2f shows the top view of the corroded surface. The corrosion attack was very uniform along the perimeter of the wire as was the thickness of the layer of degradation products. The parameters for in vitro degradation are summarized in Table 1. The thickness of the layer was approximately 2.8 µm after 28 days of immersion, which corresponds to an in vitro degraded mass Δ*m*_deg_ of 6.26 mg. The degradation depth *d*_deg_ was thus approximately 33.6 µm/y. The loss in wire diameter was identical to the degradation product layer thickness of 2.8 µm. In both images, the formation of deep cracks in the product layer is observable. The cracks formed during the drying of the samples rather than took place in-situ (see Appendix A). This is underlined by the fact that pure Mo was found in the EDX analysis between the cracks rather than a newly formed oxidic product layer.

Semiquantitative EDX analysis of the products shown in Figure 2f suggests that approximately 37–43 atomic percent (at.%) of Mo remains in the layer along with an oxygen amount of 40–45 at.%. Furthermore, there was a significant amount of elements originating from the electrolyte: approximately 2 at.% Na, 1.5 at.% Mg, 5–7 at.% P, and 7–8 at.% Ca. It is, however, of note that the measured Mo/O ratio of approximately 1:1 did not correspond to the expected Mo oxides MoO_2_, Mo_2_O_5_, or MoO_3_ [16,18], which may be explained by the semiquantitative nature of the executed EDX analysis that is limited in the detection of light elements like oxygen. An EDX analysis of the areas between the cracks showed 100 at.% Mo, which is another indicator that the product layer cracks as a whole upon drying.

Overall and daily mass losses due to dissolution of the Mo wire as derived from the measured Mo ion concentrations in c-SBF–Ca are depicted in Figure 3. The dissolution mass loss and dissolution rate after 28 days of immersion are summarized in Table 1. Note that there was a very small measurement deviation for all timepoints for the three investigated samples. The overall mass loss increased during the immersion period and reached a value of 2.57 ± 0.04 mg/cm^2^ at day 28 (Figure 3a). This corresponds to a daily mean dissolution rate of 91.8 ± 1.4 µg/cm^2^·d (Figure 3b). The measured dissolved mass and dissolution rates of Mo were only approximately 10% lower than the values for degradation. Thus, it is relatively safe to assume that only about 10 weight percent (wt.%) of the original Mo was still present in the oxides, which seems to be a more plausible value than what was measured with EDX. Furthermore, the value for the last 4-day measurement period (days 24–28) was 143.6 ± 1.1 µg/cm^2^·d, and thus much higher than the mean values. The increasing acute release of Mo was important for further biocompatibility assessments. However, in our recent study, human endothelial and smooth muscle cells showed no signs of cell damage at Mo concentrations below 15 mg/L and 120 mg/L, respectively [2], which means that no adverse effects on coronary cells in the in vivo test are expected.

The measured dissolution for cold-drawn Mo wires in c-SBF–Ca is approximately threefold higher than those of the recently investigated stress-relieved Mo sheet material [2]. This increase is most likely related to the different microstructure of the wire. The high concentration of dislocations due to the high degree of deformation may accelerate degradation. The approximate degradation rate of 33.6 µm/y is higher than the benchmark of 20 µm/y for a bioresorbable stent material [7]. Since degradation and dissolution accelerate over time, even higher rates are to be expected in the long term. Furthermore, due to the high mechanical strength of Mo, smaller stent struts of 50 µm or less might be possible, decreasing the benchmark value for total degradation to below 2 years. In conclusion, the metallurgical and geometrical state seems to be a major influence on the degradation rate. This could be used to tailor degradation of Mo-based resorbable implants in the future.

### 3.2. In Vivo Rat Model

#### 3.2.1. X-ray and CT Visibility

After 3, 6, and 12 months, implanted Mo wires were clearly detectable by means of radiography and CT (Figure 4). There was no discernible difference in visibility between the implanted wires and the reference wires outside the rat body. Radiopacity is one of the essential requirements for stent materials since stent placement is guided by radiographic imaging. Thus, the high radiopacity of Mo due to its high density and atomic number may be advantageous compared to clinically used stent materials such as nitinol or the resorbable Mg-based alloy used in Magmaris^®^ stents, where radiopaque markers must be used.

#### 3.2.2. In Vivo Corrosion Behavior in the Abdominal Rat Aorta

All the samples explanted after 3, 6, and 12 months showed signs of degradation in the form of dark discoloration (see Figure 1a). The appearance is similar to the samples corroded in vitro (Figure 2a). However, the degree of degradation was very different for the individual samples explanted at the same timepoint. Therefore, the influence of implant location on degradation is discussed first by the example of three samples explanted after 12 months, where the differences could be most clearly discerned. Figure 5 shows photographs and cross-sections of the samples. The cut-out segments of the descending aorta housing the implant wire are shown in Figure 5a,d,g. The wires in Figure 5a,d were only partially integrated into the vessel wall. Part of the wire protruded into the abdomen; the other half pierced through the wall into the lumen. This dislocation of the wires in the vessel could have occurred either after implantation or, more likely, during aorta explantation. In contrast, the wire in Figure 5g remained in its proper location inside the vessel wall. All the wires were colored black, which indicates at least some degree of degradation (compare to Figure 1a). Longitudinal dissection of the vessel would reveal the other side of the implanted wire (Figure 5b,e,h). One sample protruded into the lumen (Figure 5b), the second was close to the vessel wall and embedded in a thin layer of tissue (Figure 5e), and the third was completely integrated into the vessel wall (Figure 5h).

Figure 5c,f,i shows the cross-sections indicated by the red lines in Figure 5b,e,h. A uniform product layer formed around each wire. No locally enhanced corrosion was observed for these wires and all the other implanted wires. The diameter of the metallic part of the wire decreased to less than the original 250 µm. As seen in Figure 5c, the diameter of the wire section protruding into the lumen only decreased by 4 µm after 12 months (Figure 5c). The thickness of the degradation product layer was 2 µm. The diameter of the wire in Figure 5f decreased by 15 µm. The thickness of the degradation product layer was 6.5 µm in that case. The wire in Figure 5i that was completely embedded in the vessel wall showed the most severe degradation, with a loss in metallic diameter of 27 µm and a product layer that was 11 µm thick.

Consequently, the better the embedding and contact to the vessel wall tissue, the stronger was the degradation of Mo. This finding is analogous to the results reported by Pierson et al. for Fe wires [24]. Furthermore, the following discussion of the degradation behavior, the product layers, and the suitability of Mo for application in cardiovascular stents focuses mostly on the wires properly embedded in the vessel walls. These wires are in an environment closest to the expected environment of an implanted cardiovascular stent.

Figure 6 shows cross-sectional and surface SEM images of explanted Mo wire samples after 3, 6, and 12 months of implantation. A summary of the degradation layer thickness for all the investigated explanted wire samples is shown in Appendix A. The images show areas of the wires that were properly embedded in the arterial vessel wall. After 3 months of implantation, a degradation product layer of less than 1 µm in thickness was formed (Figure 6a,b). No local corrosion attack was observed. There was no cracking of the layer observed due to low thickness. The EDX analysis of the surface revealed 48–52 at.% Mo and 40–43 at.% O. The relatively high amount of Mo compared to 6 or 12 months was most probably due to partial excitation of the metallic Mo below the thin layer. Furthermore, approximately 3 at.% Na, 2 at.% P, and 4 at.% Ca were detected.

After 6 months, the product layer for the sample was clearly visible and had a thickness of 4.1 µm (Figure 6c,d). The second sample (not shown) that was completely covered with the vessel wall tissue had product layer thickness of 5.7 µm. The thickness of the layer was very uniform, and no signs of a local corrosion attack were observed. Surface analysis again showed 41–43 at.% Mo and 42 at.% O, as well as 3 at.% Na, 4 at.% P, and 8 at.% Ca. The composition was very similar to that of the degradation products found in vitro.

Only two wires from the 12-month cohort were completely integrated into the vessel wall. Still, the degradation was very uniform, with degradation product layers of 11.1 µm and 11.0 µm, and no localized corrosion attack was observed. EDX analyses for the cross-section and the surface of the product layers yielded similar elemental compositions. Mo values ranged between 31 and 46 at.% while the O values ranged between 25 and 44 at.%. Furthermore, 2–5 at.% Na and 1–2 at.% Mg were found on the surface. The amount of P and Ca increased from 6 to 12 months of implantation, with the values of 7–13 at.% and 11–17 at.%, respectively. This may be the first evidence that dissolving Mo oxides are gradually replaced by calcium phosphates.

The morphology and (semiquantitative) composition of the degradation products formed in vitro and in vivo were highly similar (see Figure 2). Thus, we conjecture that the degradation mechanism described in our previous study [9], which was derived mostly from the works of Johnson et al. [17] and Petrova et al. [18], also applies to the in vivo environment. Furthermore, we assume that the cracks in the degradation product layer of the samples explanted after 6 and 12 months formed during cleaning and drying of the samples after explantation rather than in-situ because the same was observed in vitro (see Appendix A) and there was no localized corrosion in the vicinity of the cracks.

In vivo degradation can be assessed analogously to the in vitro analysis based on the volume of the product layer. Due to the uniformity of the product layers at all the timepoints, the volume could be calculated from the product layer thickness and the total length of the Mo wire, assuming that the entire length of the wire was embedded in the vessel wall tissue and therefore situated in the same physiological environment. The calculated degraded volume and mass as well as the degradation and dissolution rates are summarized in Table 2:

There was an increase in degradation rates up to 33 µg/(cm^2^·d) in the first 6 months, which remained nearly constant afterwards. However, this finding cannot be statistically verified due to the low number of samples that is considered properly evaluable and should be understood as the first rough estimation. For comparison, a hypothetical stent with a weight of approximately 15 mg, a strut with the thickness of 50 µm and square strut profile has a surface area of 1.2 cm^2^ and a volume of 1.5 mm^3^. Considering the derived degradation rate, it would be completely degraded in approximately one year. On the other hand, the degradation depth of 13.5 µm/y for the 12 months timepoint is lower than the corresponding 25 µm/y (50 µm struts degrading in one year). This shows that the latter value is not a good indicator for the degradation assessment of small-profile implants like stents. However, in both cases, the gradual reduction of the metallic surface area during degradation was not considered.

Quantifying dissolution in vivo is more complicated than in vitro since the mass loss in vivo cannot be measured directly by means of ICP-OES. Based on the similarity of the product layers in vitro and in vivo, we can assume that the Mo content of the product layer in vivo is around 40 at.% at the most. Based on the volume of degradation products, this corresponds to approximately 0.92 mg of dissolved Mo after 12 months of implantation. Based on the original surface of the wires (length, 15 mm, ø0.25 mm), a total dissolution of 7.8 mg/cm^2^ and a dissolution rate of 22 µg/(cm^2^·d) could be derived. The dissolution rate was lower than the dissolution observed in vitro after 28 days of immersion in c-SBF–Ca by a factor of four. Those figures give an indication of the potential magnitude of accumulation of dissolved Mo in the liver, kidneys, and other organs for the following biocompatibility assessment.

#### 3.2.3. Mo Concentrations in Serum and Urine

Serum concentration of Mo both in animals with implanted wires and the control group was below the detection limit of the ICP-OES device of <1 ng/mL. This is in accordance with the findings for the control group of Sprague–Dawley rats investigated by Murray et al. [27] where almost no Mo was detectable in serum. Human serum concentrations are also well below 1 ng/mL in healthy persons with a normal diet [28].

Since urine was sampled at different timepoints for individual animals and was not collected over a certain period in this study, the measured urine concentrations of molybdenum strongly varied for individual animals in the implant group from 5 to 160 ng/mL. In the control group, the values were between 55 and 164 ng/mL. Thus, there was no observable increase in Mo concentrations. This is not surprising since a maximum of <1 mg of Mo was dissolved from the implanted wires over a period of 12 months.

#### 3.2.4. Blood Analyses and Inflammation

Blood cell populations of the animals with implanted wires and the age-matched control rats were analyzed and compared (Figure 7). While all the measured cell populations were found within a physiological range, we measured lower amounts of red and white blood cells as well as of platelets at the timepoints of 3 and 6 months compared to the control group (Figure 7a–c). After 12 months, these differences were no longer detectable. Further differentiation of the white blood cell population revealed that 3 months after wire implantation, the number of neutrophils was reduced, while after 6 months, the monocyte population was diminished.

Molybdenum is an essential trace element, which, as an enzyme component, supports iron metabolism and thus contributes to hematopoiesis. Due to the low measured Mo concentration that reaches the periphery via degradation of the wire, a direct influence on diminished hematopoiesis or blood cell degradation is unlikely [29]. This is in accordance with the findings of Murray et al. who reported no effects on hematological parameters after 90 days of sodium molybdate dihydrate feeding of Sprague–Dawley rats [30]. Contrasting these findings, Pandey et al. investigated metal accumulation in the tissues of male rats who orally received 50 mg molybdenum per kg in eight consecutive doses and found significant Mo accumulation in blood cells, which they concluded as a possible risk for anemia [31]. However, the dose given in that study was much higher than the Mo released at the low degradation rates in our study. Thus, it is conceivable that the reduced blood cell values observed after 3 and 6 months were due to the injury of the vascular system during the surgical implantation of the wires. This is supported by the equalization of the amount of red and white blood cells and platelets after 12 months. Finally, blood cell populations in rats can fluctuate strongly even without a pathological cause [32].

The levels of C-reactive protein revealed to be below the detection limit of 0.6 mg/l in all the groups, indicating no inflammatory response to the degradation of the Mo wire.

#### 3.2.5. Mo Concentrations in Organs

Figure 8 summarizes the ICP-OES-derived Mo concentrations in the abdominal aorta, kidneys, and liver for the 3, 6, and 12 months timepoints. As expected, Mo concentrations were elevated in the direct vicinity of the implant in the abdominal aortic vessel wall. In contrast, Mo concentrations in the control group were below the detection limit. For the animals with implanted wires, concentrations between 0 and 570 µg/g tissue were measured. The mean values were 57 µg/g, 210 µg/g, and 65 µg/g for the 3, 6, and 12 months groups, respectively. Nevertheless, future studies will have to show if these increased Mo concentrations in the vessel tissues result in histological and, even more importantly, physiological alterations.

There were very large deviations between the measured values for individual animals that did not correspond to the manner of implantation, i.e., whether the wires were properly situated in the vessel wall or not. The mean mass of the examined aortic segment was 16.8 ± 3.2 mg, 12.7 ± 3.5 mg, and 11.6 ± 2.7 mg for the 3, 6, and 12 months cohorts while the total amount of Mo was 0.8 µg, 2.8 µg, and 0.8 µg, respectively. This is far below the estimated amounts of dissolved Mo from the implanted wires from paragraph 3.2.2 of 40 µg, 280 µg, and 760 µg, respectively. This is the first indication that dissolved Mo is at least transported away from the implant site.

The Mo concentrations in the kidney and liver tissues were very low. In both the implant and the control groups, Mo concentrations were below 0.1 µg/g in the kidneys for the 3 and 6 months cohorts and approximately 0.9 µg/g for the 12 months cohort. The latter value is in good accordance with the amounts in the control group reported by Murray et al. [27]. The increase for the 12 months cohort may be attributable to ageing of the rats. Considering the total mass of a rat kidney of approximately 1 g in female rats [27], the total amount of Mo found was well below 1 µg for all the tested animals.

The Mo concentrations in the liver were 2–2.5 µg/g for the implant group and 1.5–2.8 µg/g for the control group, which is close to the values in the control group reported by Murray et al. [27]. The values slightly increased from the 3 months cohort to the 12 months cohort in both groups. Taking into account the total mass of a typical rat liver of approximately 11 g in female rats [27], a total incorporated Mo amount of approximately 20–30 µg can be estimated. The Mo concentration was also far below the estimated amount of dissolved Mo of 0.43 mg and 0.92 mg from the implanted wires for the 6 and 12 months cohorts. The measured Mo values for the kidneys and liver were comparable to those derived by Bersényi et al. for rabbits and by Murray et al. for rats with a high dietary Mo intake of 1.2 mg/d·kg for 14 days and 17 mg/d·kg for 90 days, respectively [30,33]. In both studies, no significant accumulation was found in the kidneys or liver, even considering the much higher dose of Mo compared to our study. In conclusion, the analysis of Mo concentrations indicates that most of the released Mo from an implant is removed from the implant site and is most likely renally excreted. Considering the results of Murray et al., it is not surprising that no adverse effects are observed at the low concentrations released from wire implants [30]. This is still true for a hypothetical Mo stent with an approximate mass of 15 mg that degrades in one year.

#### 3.2.6. Histological Organ Analysis

Histological sections of the kidneys and liver of the Mo wire-implanted animals compared with the age-matched control animals from the 12 months cohort are shown in Figure 9. Analyses for the 3 and 6 months timepoints for the liver and kidney as well as for the spleen, brain, and lungs are found in the Appendix A. None of the organs showed any pathological changes between the study and control animals. In animal studies of rats and guinea pigs, the highest Mo tissue concentrations after oral Mo intake were measured in the kidneys [30,31,34]. Murray et al. reported no macroscopic pathological findings. Microscopically, slight diffuse hyperplasia of the proximal tubules in the kidneys of two female rats who had received the highest Mo dosage (60 mg/kg/d) was established. This finding was related to the administered Mo. In our study, neither macroscopic nor microscopic changes of the kidneys were visible, which might have been due to the comparatively low Mo concentrations in the organs. In humans, the highest amounts of Mo are also found in the kidneys and liver [35].

## 4. Conclusions

Regarding degradation behavior and biocompatibility of Mo in vivo after 3, 6, and 12 months we conclude the following:Commercially available pure molybdenum degrades in vivo.The corrosion attack was uniform at all the timepoints investigated. No evidence of a localized corrosion attack was found.The degradation products were dense and uniformly distributed on the exposed surface of the implants. The thickness of the products was approximately equal to the radial loss of the metallic part of the wire. However, there was a slight reduction in the total diameter after 6 and 12 months of implantation.After 12 months, the wires that were properly integrated in the aortic vessel walls had a degradation layer thickness of approximately 11 µm and a radial loss of 13.5 µm/y. This corresponds to a calculated degraded mass of 1.5 mg (of 7.5 mg) and a degradation rate of approximately 36 µg/cm^2^·d.Derived from semiquantitative EDX analyses, the dissolved mass of Mo was at least 60% of the degraded mass. Mo oxides were apparently gradually replaced by calcium phosphates since the amount of Ca and P increased from 3 to 12 months after implantation.The degradation and dissolution behavior was very similar for in vitro and in vivo experiments.Mo concentrations in rat serum and urine analyzed by means of ICP-OES were not increased compared to the control group.An accumulation of Mo was measured in the rat aorta walls in the direct vicinity of the implantation site. No correlation between the degradation rate or implantation quality and the amount of Mo in the vessel wall tissue was observed.Neither Mo accumulation nor histological changes in the kidneys and liver or inflammatory responses were found at all the timepoints.

In conclusion, degradation and dissolution of molybdenum were demonstrated in vivo, without the occurrence of adverse physiological effects. Molybdenum showed a unique combination of high mechanical strength, uniform and moderate degradation in vitro and in vivo, and good biocompatibility. Furthermore, the radiopacity of Mo was excellent, which is a favorable property for high visibility during implantation and surgical follow-up examinations. This makes molybdenum a very promising material for bioresorbable implant-like stents. Studies focused on long-term degradation and dissolution characteristics of Mo and animal studies with prototype Mo stents should be performed in the future.

## Figures and Tables

**Figure 1 materials-14-07776-f001:**
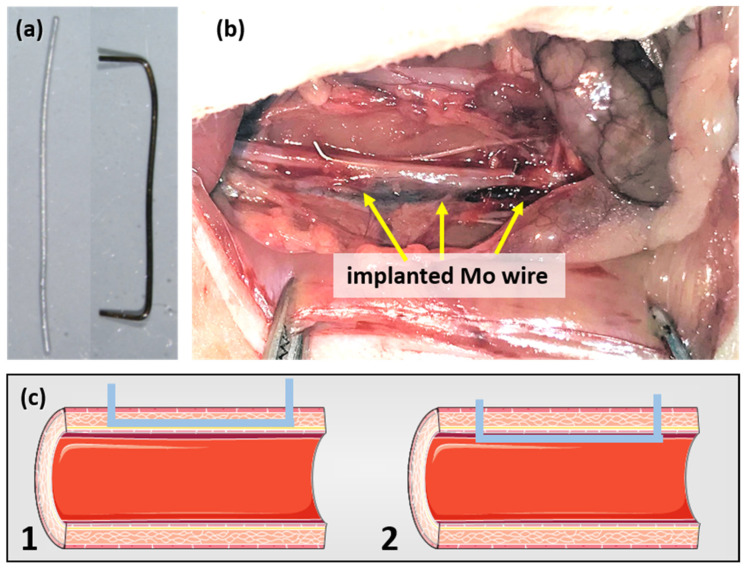
Mo wire sample before and after implantation for 3 months (**a**) and the Mo wire implanted in a rat aorta (**b**). Image (**c**) shows a schematic drawing of the position of an implanted wire in the aortic vessel wall (1) and lumen (2) (schematic images copied from smart.servier.com, accessed on 16 November 2021).

**Figure 2 materials-14-07776-f002:**
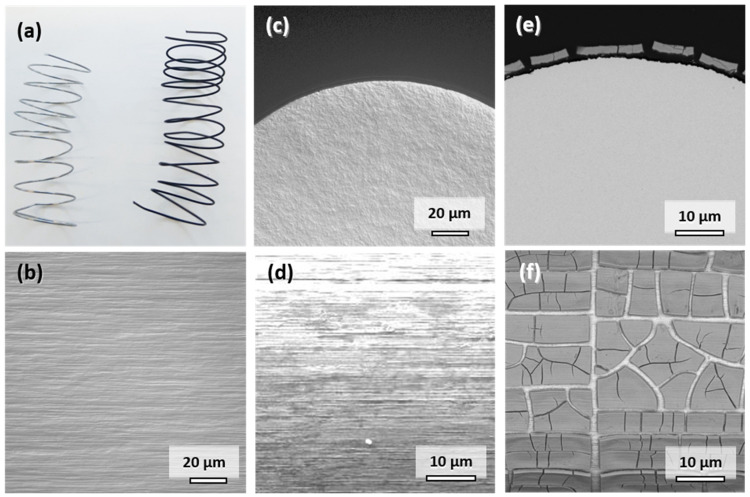
Representative photographs of the Mo wire samples before and after 28 days of corrosion in c-SBF–Ca (**a**). Optical micrographs of the microstructure in the drawing direction (**b**) and of the cross-sectional view on the manufactured Mo wire (**c**). SEM images of uncorroded surface (**d**), cross-sectional view (**e**), and top-view (**f**) after 28 days of corrosion in c-SBF–Ca.

**Figure 3 materials-14-07776-f003:**
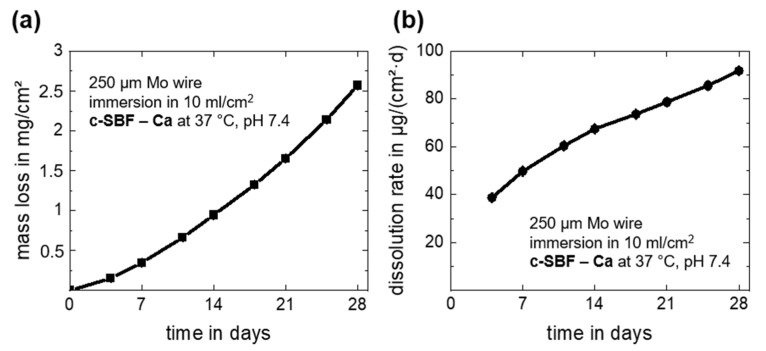
In vitro progression of the dissolution mass loss derived from ICP-OES measurements (**a**) and the dissolution rates (**b**) for the immersion of Mo wires for 28 days in c-SBF–Ca.

**Figure 4 materials-14-07776-f004:**
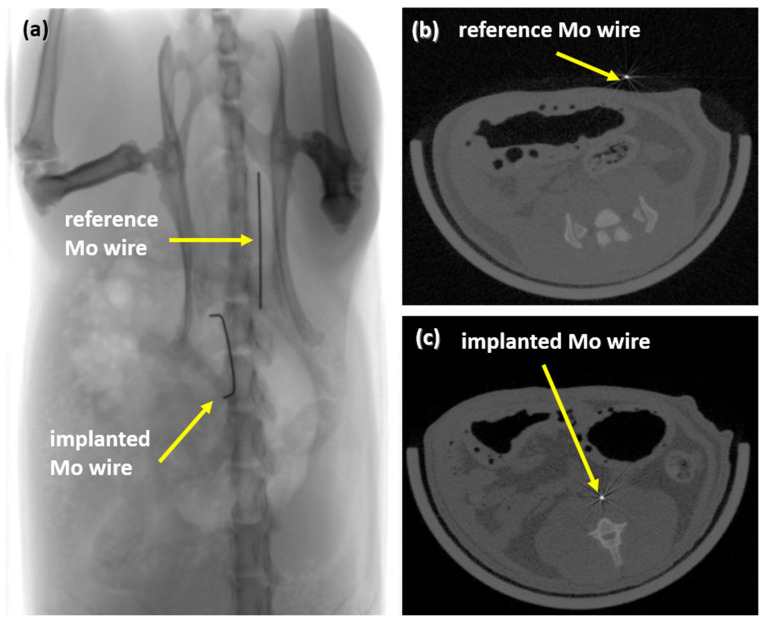
Radiograph of an implanted wire and a reference wire (**a**) and µCT images of the reference wire (**b**) and of the implanted wire in rat aortae (**c**).

**Figure 5 materials-14-07776-f005:**
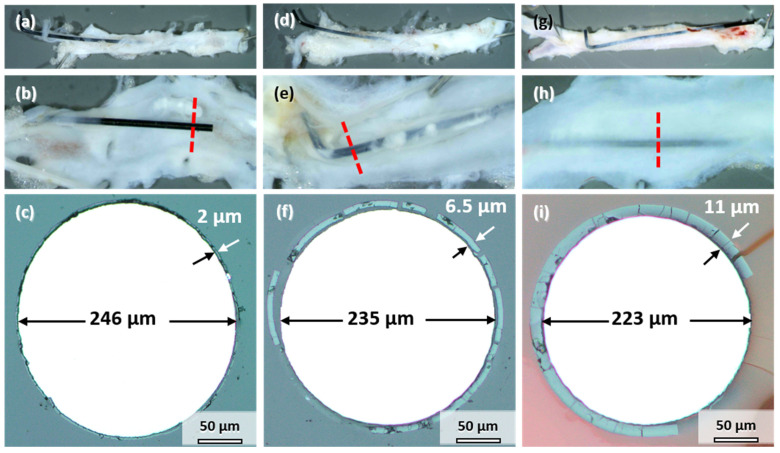
Mo wires explanted from the abdominal rat aorta (**a**,**d**,**g**), sliced open aorta (view from within) (**b**,**e**,**h**) with indicated position of the cross-sectional view of the wires (**c**,**f**,**i**) after 12 months of implantation.

**Figure 6 materials-14-07776-f006:**
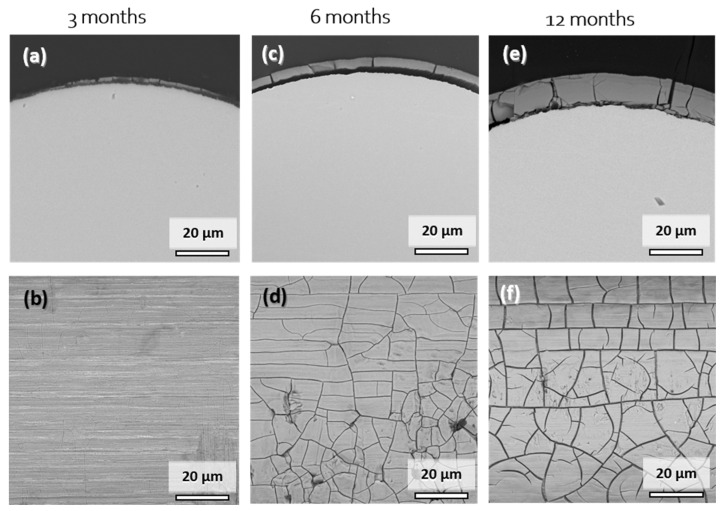
Representative cross-sectional (**a**,**c**,**e**) and top-view SEM images (**b**,**d**,**f**) of the Mo wires explanted from the abdominal rat aorta after 3, 6, and 12 months of implantation.

**Figure 7 materials-14-07776-f007:**
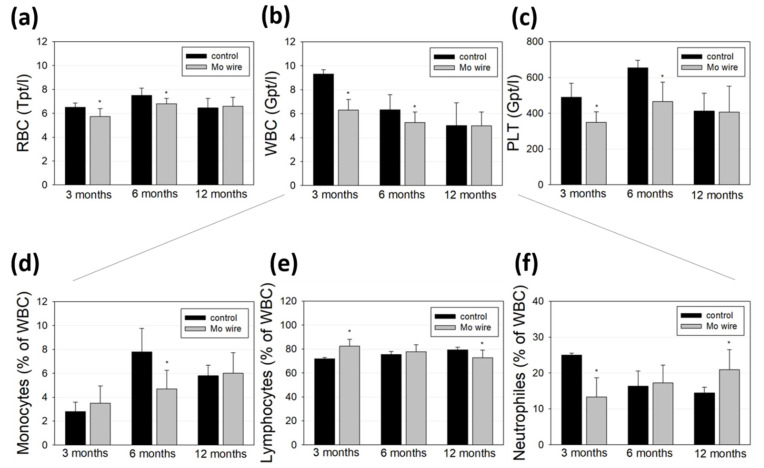
Blood analyses of the rats after 3, 6, or 12 months of Mo wire implantation compared to their age-matched controls. The bar charts represent the amount of red blood cells (RBC, **a**), white blood cells (WBC, **b**), and platelets (PLT, **c**). WBC were further analyzed for monocytes (**d**), lymphocytes (**e**), and neutrophil granulocytes (**f**); * *p* < 0.05; n = 6–14.

**Figure 8 materials-14-07776-f008:**
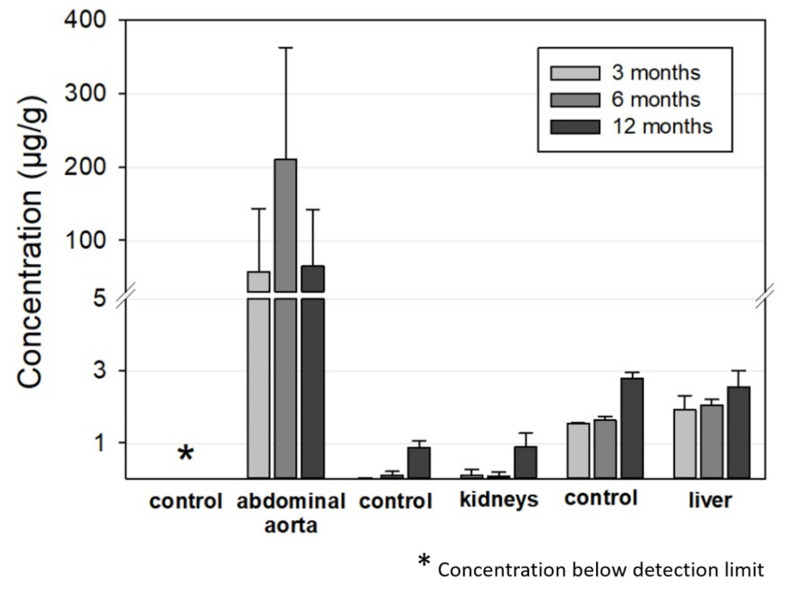
ICP-OES-derived tissue concentration of Mo in the aorta (implantation site), the kidneys, and the liver compared to the respective control organs of the age-matched animals (aorta: n_3 months_ = 13, n_6 months_ = 12, n_12 months_ = 12; liver and kidneys, respectively: n_3 months_ = 6, n_6 months_ = 7, n_12 months_ = 12; aorta, liver, and kidneys of the control animals: n = 6, respectively).

**Figure 9 materials-14-07776-f009:**
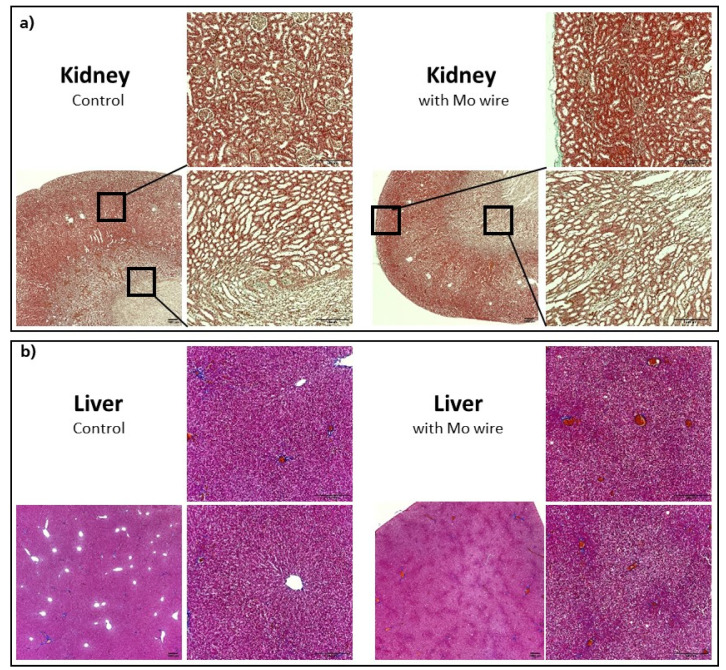
Representative histological imaging for the kidneys (Masson–Goldner staining) (**a**), liver (Azan staining) (**b**), each for the control group and the group with the implanted Mo wires for the 12 months timepoint. For the other timepoints and larger pictures, see the Appendix A.

**Table 1 materials-14-07776-t001:** Assessment of the amount of degradation of Mo immersed for 28 days in c-SBF–Ca.

Immersion Time (days)	*b*_deg_ (µm)	*V*_deg_ (mm^3^)	Δ*m*_deg_ (mg) (mg/cm^2^)	*v*_deg_ (µg/(cm^2^·d))	Δ*m*_diss_ (mg) (mg/cm^2^)	*v*_diss_ (µg/(cm^2^·d))
28	2.8	0.609	6.26 (2.85)	101.6	5.65 ± 0.09 (2.57 ± 0.04)	91.8 ± 1.4

*b*_deg_—degradation depth; *V*_deg_—degraded volume; Δ*m*_deg_—degraded mass; *v*_deg_—degradation rate; Δ*m*_diss_—dissolved mass; *v*_diss_—dissolution rate.

**Table 2 materials-14-07776-t002:** Assessment of the amount of degradation of the Mo wires implanted in abdominal rat aortae.

Implantation Period (month)	*b*_deg_ (µm0	*V*_deg_ (mm^3^)	Δ*m*_deg_ (mg) (mg/cm^2^)	*v*_deg_ (µg/(cm^2^·d))	Δ*m*_diss_* (mg) (mg/cm^2^)	*v*_diss_* (µg/(cm^2^·d))
3	1.0	0.012	0.12 (1.1)	12	0.07 (0.6)	7
6	6.0	0.069	0.71 (6.0)	33	0.43 (3.6)	20
12	13.5	0.150	1.54 (13.0)	36	0.92 (7.8)	22

*b*_deg_—degradation depth; *V*_deg_—degraded volume; Δ*m*_deg_—degraded mass; *v*_deg_—degradation rate; Δ*m*_diss_—dissolved mass; *v*_diss_—dissolution rate. * considering the minimum dissolution of 60% as derived from semiquantitative EDX analysis.

## Data Availability

The data presented in this study are available on request from the corresponding author.

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
