# Peer review of "Biocompatibility and Degradation Behavior of Molybdenum in an In Vivo Rat Model"

_materials, 2021, doi:10.3390/ma14247776_

Round 1

Reviewer 1 Report

On request of Materials, I have revised the manuscript entitled "In-Vivo Qualification of Biocompatibility and Degradation of Bioresorbable Molybdenum in a Rat Model" by Antje Schauer and co-authors. 

With this paper the authors aimed to investigate the biocompatibility and degradation behavior of pure molybdenum as bioresorbable metallic material for biomedical applications. 

Report:

The hypothesis is new although not a very original one, but the present work has a major significance for the medical and implant field. The manuscript is well written, but there are some aspects that can be improved:

  • The histology methodology should be described in more detail in the Materials and Methods section.
  • Tables 1 and 2 should also be accompanied by a legend defining abbreviations and notations even if they have been previously defined. It would make the manuscript easier to follow for a non-specialist reader.
  • From my point of view, a change in the value under the detection limit to 200 μg/g is not a slight increase. Given the accumulation of Mo in the abdominal aorta, perhaps the authors should also consider the histological changes that occur at this level. 

Reviewer 2 Report

In this manuscript, Schauer and colleagues have tested the biocompatibility and degradation of pure molybdenum as novel candidate for bioresorbable material used in implantology. The study very nicely presents animal model results which are critical for the development of implant materials and feature a new potential material for bioresorbable implant design. Although the approach is innovative and the study is interesting, some issues need to be addressed.

Major concerns:

-the variations between animals regarding the implantation is considerable, therefore the authors should stress how the conclusions of the study can be supported by the presented experiments

-the inflammatory response should be evaluated by assessing at least one of the established markers for inflammation IL-1B, IL-6, C reactive protein because it may render significant differences for implanted rats, it is possible the response to the degradation of the Mo wire

-it is not clear if the control animals were subjected to surgical procedures or not, the authors should state this and if the animals were not subjected to surgery, if so the differences observed for the treated compared to control group cannot be attributed solely to the Mo degradation processes

-page 11 rows 410-413: “Furthermore, we assume that the cracks in the degradation product layer of samples explanted after 6- and 12-months form during cleaning and drying of the samples rather than in-situ, because the same was observed in-vitro”-this should not be an assumption, the authors should have analysed the wire before the surgical procedure and the pre-implantation characteristics presented

-the methodology regarding blood sample and histology analysis is very superficially described therefore making it impossible to replicate

Minor concerns:

-the title could be more crisp and avoid “qualification” since it does not clarify the objective of the paper

-throughout the text many abbreviations are not described making it difficult for the non-specialists to follow the manuscript (eg: SBF, c-SBF-Ca, at %, m %)

-some figures could be presented as merged eg: figure 5 and 6 can be presented as one figure, as well as figures 7 and 8 which could also be merged

-it is unclear why the authors have collected faecal samples if they have not presented the data in the manuscript

-the labelling of the table presented in figure 8 is not clear

-overall written English can be improved to make the manuscript more accessible to readers

Reviewer 3 Report

Dear Authors,

the manuscript entitled "In-Vivo Qualification of Biocompatibility and Degradation of 2 Bioresorbable Molybdenum in a Rat Model" by Schauer et al. it is well done. But do I have two questions?

1. Studies show that molybdenum could have negative effects on ovarian structure and function. why has ovarian tissue not been studied?

2. Excessive molybdenum intake causes increased uric acid concentrations in the blood and urine (gout-like symptoms). why the blood chemistry parameters have not been evaluated?

Round 2

Reviewer 1 Report

The authors fulfilled all the requirements. Therefore, I recommend publishing the manuscript in its present form.

Reviewer 2 Report

The authors have addressed all the concerns I raised about the manuscript, thererfore I recommend accepting it in the prersent form.